# Opportunities for seasonal forecasting to support water management outside the tropics

Leah A. Jackson-Blake[1*], François Clayer[1], Elvira de Eyto[2], Andrew French[2,10], María Dolores Frías[3], Daniel Mercado-Bettín[4,5], Tadhg Moore[6,11], Laura Puértolas[7], Russell Poole[2], Karsten Rinke[8], Muhammed Shikhani[8], Leon van der Linden[9], Rafael Marcé[4,5]

[1]Norwegian Institute for Water Research (NIVA), 0349 Oslo, Norway
[2]Marine Institute, Furnace, F28 PF65, Ireland
[3]Grupo de Meteorología. Dpto. de Matemática Aplicada y Ciencias de la Computación, Universidad de Cantabria, Santander, Spain
[4]Catalan Institute for Water Research (ICRA), 17003 Girona, Spain
[5]Universitat de Girona, 17004 Girona, Spain
[6]Dundalk Institute of Technology, A91 K584, Dundalk, Ireland
[7]Albirem, Barcelona, Spain
[8]Department of Lake Research, Helmholtz Centre for Environmental Research, 39114 Magdeburg, Germany
[9]SA Water, Adelaide SA 5000, Australia
[10]School of Biological, Earth and Environmental Sciences, University College Cork, T23 N63K, Ireland
[11]Virginia Tech, Department of Biological Sciences & Forest Resources & Environmental Conservation, Blacksburg, Virginia, USA

*Correspondence to*: Leah Jackson-Blake (leah.jackson-blake@niva.no)

**Abstract.** Advance warning of seasonal conditions has potential to assist water management in planning and risk mitigation, with large potential social, economic and ecological benefits. In this study, we explore the value of seasonal forecasting for decision making at five case study sites located in extratropical regions. The forecasting tools used integrate seasonal climate model forecasts with freshwater impact models of catchment hydrology, lake conditions (temperature, level, chemistry and ecology) and fish migration timing, and were co-developed together with water managers. To explore the decision making value of forecasts, we carried out a qualitative assessment of: (1) how useful forecasts would have been for a problematic past season, and (2) the relevance of any "windows of opportunity" (seasons and variables where forecasts are thought to perform well) for management. Overall, water managers were optimistic about the potential for improved decision making and identified actions that could be taken based on forecasts. However, there was often a mismatch between those variables that could best be predicted and those which would be most useful for management. Reductions in forecast uncertainty and a need to develop practical hands-on experience were identified as key requirements before forecasts would be used in operational decision making. Seasonal climate forecasts provided little added value to freshwater forecasts in these extratropical study sites, and we discuss the conditions under which seasonal climate forecasts with only limited skill are most likely to be worth incorporating into freshwater forecasting workflows.

## 1. Introduction

We rely on freshwaters to deliver a range of vital services, and managing catchments and lakes to ensure these services are delivered can be highly challenging. Unexpected seasonal climate conditions can exacerbate the problem, as heatwaves, droughts or prolonged wet periods can stress already vulnerable systems. Advance warning of flow, water quality or biological conditions, a season in advance, could pave the way for protective measures to be put in place, with potentially great ecological, economic and societal benefits (Bruno Soares et al., 2018; Bruno Soares and Dessai, 2016). Seasonal

forecasts have obvious potential to assist management of flow-regulated catchments, where water level can be adjusted in anticipation of wet or dry seasons, and much attention has been given to this in recent years (e.g. Maurer and Lettenmaier, 2004; Turner et al., 2017; Turner et al., 2020; Peñuela et al., 2020). However, there are many other situations where forecasts could assist water managers in delivering key services, protecting vulnerable aquatic habitats and species, and in meeting environmental objectives.

Within the water sector, predicting conditions a season in advance can make use of two sources of seasonal predictability: (1) antecedent and initial conditions, for example how much water is stored in the catchment/lake at the start of the period; and (2) how weather is likely to evolve over the coming season. The relative importance of these varies greatly by location and depends on the catchment/lake characteristics, season, forecast horizon and variable of interest (e.g. Shukla and Lettenmaier, 2011; Arnal et al., 2018). To incorporate both sources of predictability into forecasts, seasonal climate model

output can be used to drive statistical or process-based surface water models. Seasonal climate models provide, for instance, probabilities of wetter or drier, cooler or hotter conditions several months in advance. One of the main sources of seasonal climate predictability is the coupled ocean–atmosphere El Niño/ La Niña pattern (Troccoli, 2010), so seasonal climate models tend to perform better in the tropics, which are more affected by these phenomena  (e.g. Manzanas et al., 2014; Beverley et al., 2019; Johnson et al., 2019). Away from the tropics, seasonal climate forecasting is challenging, and forecast

quality varies geographically and strongly depends on the variable and season of interest. The added value of using seasonal climate forecasts in freshwater forecasting outside the tropics is therefore often less clear (e.g. Peñuela et al., 2020; Arnal et al., 2018).

Many seasonal hydrologic and drought prediction systems have been developed over the last decade using a variety of forecasting methods. Seasonal streamflow forecasting is the most advanced, with many examples of systems that produce

regional or even global operational forecasts (e.g. Arnal et al., 2018; Bennett et al., 2017; Emerton et al., 2018; Prudhomme et al., 2017; Wood and Lettenmaier, 2006). These systems generally perform better than climatology (i.e. resampled historic streamflow), although forecasting methods that incorporate seasonal climate data have only been found to perform better than methods that rely on historic meteorological data over shorter lead times and/or in certain locations and seasons (e.g. in winter for 40% of Europe; Arnal et al., 2018). For lake water level, probably the longest-established operational seasonal

forecasting system is for the Great Lakes in the USA/Canada, where empirical and process-based catchment and lake models

are forced with historic meteorological data, in some cases taking monthly climate forecasts and long-term climate projections into account by the use of weightings (Gronewold et al., 2011; Gronewold et al., 2017; Fry et al., 2020). This system has been shown to offer some skill, although forecasted variability is generally lower than observed (Fry et al., 2020). Seasonal forecasts of water quality and ecology are meanwhile rare, despite their potential relevance for management. The few examples we could find included river nutrient loads in a Korean catchment (Cho et al., 2016) and turbidity exceedance in a drinking water source in the Pacific Northwest (Towler et al., 2010), and while both studies focused primarily on method development, Towler et al. (2010) showed that their workflow, which incorporated seasonal climate forecasts, resulted in an improvement in skill over climatology.

For standing waters, the use of short-term weather forecasts, i.e. timescales of up to 10 days ahead, has been advanced in a number of lake water quality studies (e.g. Thomas et al., 2020; Carey et al., 2021), but seasonal time-scales have not been addressed to our knowledge. The focus of the WATExR project, a European Union (EU) project funded by the European Research Area for Climate Services (ERA4CS), was therefore to help address this gap, by focusing on seasonal forecasting of water quality and ecology and including standing waters. Pilot seasonal forecasting tools were co-developed with water managers at five catchment-lake case study sites, four in Europe and one in South Australia. The focus was on extratropical areas, where seasonal climate predictability is lower. Tools link seasonal climate forecasts with models which predict freshwater variables of interest to decision makers at each site, including river discharge, lake water level and water temperature (described in detail in Mercado-Bettín et al., 2021), water quality, algal bloom risk and fish migration.

The substantial advances in the development of operational streamflow forecasting systems have enabled improved water management in some areas of the world. In a recent study, for example, Turner et al. (2020) found that a large proportion of dams and reservoirs in the US use seasonal stream inflow forecasting to inform water release. However, snowpack data was inferred to be the main source of information for deriving streamflow forecasts, with more limited evidence for seasonal climate information being used. Certainly in Europe, recent studies have found that seasonal climate products are still rarely used to inform water management (e.g. Bruno Soares et al., 2018). Barriers to use include low climate forecast skill at extratropical latitudes and the probabilistic nature of the forecasts, as well as factors such as a lack of awareness of what is available, accessibility, and level of expertise or training required (Bruno Soares and Dessai, 2016; Bolson et al., 2013). A variety of studies have emphasized that a key way of increasing the use of climate products in decision making is co-development, whereby scientists and decision-makers together frame and develop the scientific information and tools that are useful and usable for decision-making (Brasseur and Gallardo, 2016; Bruno Soares and Dessai, 2016). Another key aim of the WATExR project was therefore to facilitate and explore the value of using seasonal climate information to help support freshwater management. A case study-based approach, and involving water managers through every stage of development, ensured that the forecasting tools developed were user-friendly and tailored to individual site needs.

In this paper, our main aim is to test how useful the forecasting tools developed as part of the WATExR project are for supporting decision making. To do this, we first used the forecasting tools to simulate historic seasons at the case study sites and then assessed, together with end users, the potential for improved management and key challenges. This assessment process involved two exercises. In the first (Sect. 3.1), we generated forecasts for a single historic season, selected by water managers, when seasonal climate resulted in problematic conditions in each study site. Managers then assessed how useful forecasts would have been, whether they would have helped mitigate the impacts of the seasonal event, and identified barriers to operational use. In the second exercise (Sect. 3.2), we carried out a more comprehensive assessment of the seasonal forecasting windows of opportunity at each site, i.e. those seasons/variables/event types which could be reliably forecasted, their perceived usefulness, and which windows of opportunity would be of most use for management. We then discuss results in terms of the wider literature and review the opportunities and barriers for seasonal forecasting to support water management (Sect. 4). This includes a discussion of the conditions under which seasonal forecasting is most likely to be useful for decision-making, where the use of seasonal climate forecasts is most likely to provide added value, and future priorities.

## 2. Methods

### 2.1. Case study sites

Forecasting tools were developed at five case study sites, four in Europe and one in South Australia (Fig. 1). All are catchment-lake systems, and at all but the Irish site the lake water level is regulated. The main characteristics of the catchments and lakes/reservoirs are given in Table 1 and catchment maps are shown in SI1).

Mount Bold Reservoir is the largest in South Australia. Its main water supply is the Onkaparinga river, but during the dry season inflows are supplemented with pumped water from the Murray River via a pipeline. Mount Bold provides water to the Happy Valley reservoir further downstream, which is the drinking water source for the city of Adelaide (around 1.3 million people). Pumping water through the Murray pipeline is expensive, and operational decisions relating to pumping and release would benefit from advance knowledge of the likely hydrological conditions. In addition, the reservoir is susceptible to phosphorus resuspension which contributes to algal blooms in Happy Valley reservoir.

The Wupper reservoir is located near Cologne and is created by a large dam on the Wupper River. Its catchment is heavily built up compared to the other sites and the reservoir is an important recreational area. The water level is managed for flood control, maintaining environmental flows and recreation, all of which are challenging as water level fluctuations are large. Cyanobacterial blooms are problematic during hot summers and low water levels.

Lake Vansjø is located in one of the most agricultural areas of Norway which, combined with the prevalence of phosphorus-rich clay soils in its catchment, mean it is prone to poor water quality. The lake has two main basins, an eastern and a western basin, and is a very important recreational area. The eastern basin (Storefjorden) is deeper and provides drinking

water to the city of Moss (population 60,000), and flows in to the shallower western basin (Vanemfjorden). The focus in this study was on the western basin, where algal blooms are particularly problematic. Toxic cyanobacteria blooms led to bathing bans in much of the period 2000-2007, for example. Water level in the lake is regulated for hydropower, flood protection and recreation.

Sau reservoir is the main water supply source for the Barcelona metropolitan area, with a population of up to 4.4 million. It is part of the Ter River catchment, which is the main source of water to the reservoir. The reservoir is vulnerable to both wet and dry seasonal climate events, as high river discharge washes in nutrients from the catchment resulting in poor water quality, whilst dry and warm seasons may result in low water levels, algal bloom development and anoxia.

Sau, Mount Bold and Wupper Reservoirs are part of a larger chain of reservoirs, and water managers therefore face challenges in developing optimum release and pumping strategies throughout the chain. All the lake and reservoir sites face water quality challenges. High river discharge may wash in excess nutrients and lead to poor water quality, whilst prolonged dry and warm periods are often associated with algal blooms. The primary management opportunity at these sites is therefore adjusting water storage, release and pumping strategies to minimise operational costs, whilst ensuring drinking water provision, flood protection, recreation, maintaining minimum environmental flows and meeting environmental water quality targets. Advance warning of cyanobacteria bloom risk was of particular interest to the end user in Lake Vansjø, to inform lake monitoring strategies, as well as the likelihood of meeting water quality environmental targets. Additional background information on these four lake/reservoir sites is provided in Mercado-Bettín et al. (2021).

In the Burrishoole catchment in northwest Ireland the focus was on the timing of diadromous fish migration. This site is an extremely important Atlantic salmon and eel research catchment, with historic data on fish migration since the 1950s together with a comprehensive catchment monitoring programme. The primary end user interest at this site was therefore sustainable management of diadromous fish stocks, and the development of a prototype seasonal forecasting tool that could be potentially transferred elsewhere in Ireland, for example to inform the timing of eel trap and release schemes, which are carried out in Ireland to enable eels to safely migrate around run-of-river hydropower structures.

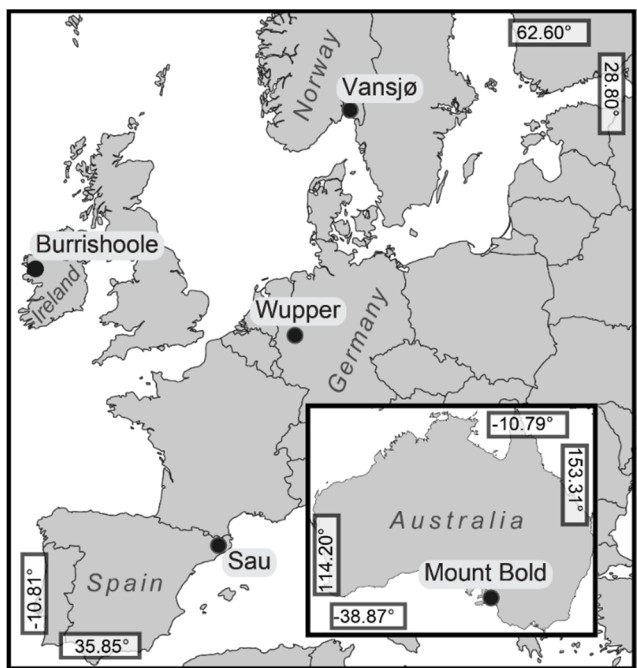

**Figure 1.** Location of the five study sites in Europe and Australia.

**Table 1. Main characteristics of the catchment-lake study sites and the water managers involved in the project.**

| Site | End user | Catchment area (km²) | Lake area (km²) | Land use (%) | Lake elevation (m) | Max depth (m) | Water residence time (yrs) |
|---|---|---|---|---|---|---|---|
| Mount Bold reservoir, Australia | South Australian Water (Senior research program manager, environmental release manager) | 357 | 2.5 | Pasture: 39 Semi-natural: 19 Built-up: 18 Crops & horticulture: 14 Other: 10 | 244 | 45 | 0.2 – 0.6 |
| Wupper reservoir, Germany | Wupperverband (Head of water resources and flood risk management, technical staff) | 215 | 2.1 | Grassland: 48 Built up: 25 Forest: 19 Agriculture: 8 | 250 | 31 | 0.2 |
| Burrishoole catchment, Ireland | Marine Institute (Station manager, postdoc) | 85 | 3.9 | Rough grazing: 73 Forestry: 22 Waterbodies: 5 | 25* | N/A | N/A |
| Western basin of Lake Vansjø, Norway | Morsa river basin management authority (Manager) | 690 | 36 | Forest: 79 Agriculture: 16 Waterbodies: 5 | 26 | 19 | 0.21 |
| Sau reservoir, Spain | ATL Water Supply Company (Head of water treatment and resource, technical staff) | 1522 | 5.7 | Forest, semi-natural: 83 Agriculture: 15 Built up: 2 | 425 | 60 | 0.2 |

* At Lough Feeagh, around 100 m upstream of the fish monitoring point.

## 2.2.    Co-development and assessment

The water managers engaged in the project included reservoir operators and water supply companies in Mount Bold, Wupper and Sau reservoirs; the Morsa river basin management authority in Norway, a partnership organisation responsible for implementation of the European *Water Framework Directive* (WFD; 2000/60/EC) in the catchment; and the Marine Institute in Ireland, who are responsible for diadromous fish stock monitoring and informing fisheries management (Table 1). One or two people were involved in the project from each organisation, including someone in a position to take water management decisions or, in the case of Mount Bold, responsible for research into new operational methods. A technical staff member (i.e. with responsibility for running models and analysing data) from each end user organisation was also involved the project, where necessary. Water managers from Mount Bold and Wupperverband had experience in using catchment hydrology, water quality and reservoir models, and the ultimate goal at these sites, would be to run seasonal forecasting workflows in-house. At Sau and Vansjø, the aim was to develop user-friendly forecast output, rather than for the end users to run the workflows themselves.

Water managers were involved in the design of the forecasting tools from the start as active project members, to ensure the tools matched their interests and needs. This also meant that they were able to interpret the probabilistic forecasts and the reliability information included with forecasts, and so carry out an informed assessment of the value of forecasts for decision making. Formal co-development and assessment exercises included:

- An initial workshop to introduce seasonal forecasting and define the main management challenges and priorities at each site and ways in which forecasts could contribute to decision making;
- A forecasting tool co-development workshop to agree on desired features, functionality and information layout at each site;
- A workshop on communicating and visualising seasonal forecast uncertainty and reliability information (more details and outcomes are briefly described in Sect. 2.4);
- Two interactions to assess end user perceptions on the qualitative value of forecasts, i.e. the practical potential for improved management (the focus of this paper):
  1. *Assessment of the usefulness of forecasts for a selected historic event* (Sect. 3.1): this involved water managers first selecting a historic season of interest. Researchers then generated forecasts for this season and shared them with managers, who were asked a set of questions via an on-line questionnaire to determine their interpretation of the forecasts and their potential usefulness (full questions are given in SI2; see Sect. 6). Results were then discussed individually between researchers and water managers at each case study site via a facilitated virtual call, and then case studies shared experiences and main findings at an all-hands workshop.
  2. *Assessment of the usefulness of windows of opportunity* (Sect. 3.2): researchers at each case study site generated a list of the variables/seasons/event types that could be reliably forecast (the windows of opportunity). The potential value of these for management was then explored via an on-line survey, where water managers were also asked to

select any additional windows they were most interested in obtaining reliable forecasts for (see Box A1 in the
190 Appendix for survey design).

## 2.3. Forecasting workflows

The surface water variables of interest varied between sites. The Irish site focused on the timing of diadromous fish migration to inform fisheries monitoring and management. Elsewhere, all water managers were interested in streamflow forecasts. Water temperature was also of broad interest, as the most basic water quality parameter which affects a host of
195 other biogeochemical and ecological processes, and which can be simulated relatively robustly using process-based models. This was therefore the "water quality" end point forecast in the German, Spanish and Australian sites. At lake Vansjø in Norway, water temperature alone would not be enough to inform decision making, and the end user was also eager for forecasts of water quality parameters, in particular the risk of toxic cyanobacteria blooms.

A range of different models were used to produce forecasts for the freshwater variables of interest (hereafter termed
freshwater "impact models"). In most cases, these impact models integrate seasonal climate forecasts, knowledge of antecedent conditions and the characteristics of the system to predict the future state. The following models were used to simulate the different variables of interest:

- **Streamflow, lake level and lake water temperature**: Details of the modelling workflow used in the lake/reservoir sites are given in Mercado-Bettín et al. (2021). In brief, a process-based catchment hydrology model was used to simulate
streamflow, and in turn provided input to a process-based lake model which simulated lake water level and temperature. Differences in site characteristics led to different models being used at different sites (Table 1). For catchment hydrology, the spatially distributed mesoscale Hydrologic Model (mHM, www.ufz.de/mhm) was used at Sau, the semi-distributed SimplyQ model (Jackson-Blake et al., 2017) at Vansjø, and the semi-distributed *Genie Rural* (GR) models, implemented within the *airGR* R package (Coron et al., 2017), were used elsewhere (GR4J at Mount Bold and Burrishoole, GR6J at
Wupper). Lake thermodynamics and water level were simulated using GOTM (http://gotm.net) in Vansjø and Sau and GLM (Hipsey et al., 2019) in Mount Bold and Wupper reservoir.

- **Timing of fish migration:** At the Irish site, a statistical model was developed to predict the timing of seawards migration of Atlantic salmon (*Salmo salar*), brown trout (*Salmo trutta*) and European eel (*Anguilla anguilla*). Daily fish counts were estimated for each species using correlative models, with predictor variables stream discharge, water temperature, a
215 proxy for fish preparedness for migration, moonlight exposure and, for eels, rate of change in water temperature over the previous 20 days. Daily stream discharge was estimated using GR4J. Daily water temperature was estimated using a four parameter air temperature to water temperature statistical model, where daily water temperature was linearly correlated with lagged air temperature. Fish preparedness for migration was estimated by first estimating the photoperiod-weighted degree days after the winter solstice (taking as input photoperiod and water temperature data), and then fitting fish count
data to non-linear unimodal functions of photoperiod-weighted degree days.

- **Algal bloom risk:** At the Norwegian site, algal bloom risk was estimated using a continuous Gaussian Bayesian Network (BN). Water quality observations from the previous year were used to produce probabilistic estimates for growing season (May-October) mean concentrations of total phosphorus (TP), chlorophyll-a (chl-a) and lake colour, and growing season maximum cyanobacteria biovolume (cyano), incorporating interrelationships between variables. Meteorological nodes were not included in the network, after cross-validation showed that they did not increase (and sometimes decreased) predictive performance, meaning that this impact model did not include seasonal climate data as input.

To produce seasonal surface water forecasts, freshwater impact models were first 'warmed up' where necessary using historic meteorological forcing data, and then run for the future target season of interest using seasonal climate model output as forcing data.

For historic meteorological data, we used the ERA5 reanalysis data, the latest reanalysis produced by the European Centre for Medium-Range Weather Forecasts (ECMWF; Hersbach et al., 2020), which is a global dataset with 0.25° horizontal resolution and hourly temporal resolution. For seasonal climate predictions, we used the ECMWF's most recent long-range forecasting system SEAS5 (Johnson et al., 2019), a global dataset with 1° horizontal resolution. SEAS5 seasonal forecasts are available as real-time operational forecasts from 2018 (50 members), and as retrospective seasonal forecasts for past years (hindcasts) for the period 1993-2016 (25 members), which were used in this study. SEAS5 was bias corrected using ERA5 using quantile mapping (see Mercado-Bettín et al., 2021 for details). All climate data were downloaded and post-processed using the climate4R bundle of packages (Iturbide et al., 2019). ERA5 is a natural choice to use in combination with SEAS5 as it is used to initialise SEAS5 and therefore ensures consistency between variables. Local sources of meteorological data could be used instead, and may be less biased than ERA5. However, ERA5 is available on a global grid and for the period of time, variables and temporal resolution required in the project, which was not the case for local data in a number of the case studies. A workflow using ERA5 is also more easily generalisable and transferable, which was one of the project's objectives. However, in the Irish site local meteorological data was used to bias correct the ERA5 data, and in future workflows it would be worth exploring the more recent bias-corrected ERA5 data (Cucchi et al., 2020). Alternative seasonal climate forecasting systems were used at the start of the project, before SEAS5 became available (CFS, SEAS4), without substantial differences in seasonal climate forecasting skill to SEAS5.

Forecasts of catchment discharge and lake temperature were produced four times a year for the boreal seasons spring (March-May), summer (June-Aug), autumn (Sep-Nov) and winter (Dec-Feb). The fish model and BN produce one forecast per year, for the months when seaward fish migration occurs in Ireland and the 6-month (May-October) 'growing season' used in Water Framework Directive (WFD) ecological status classification in Norwegian lakes.

**2.4.        What does a seasonal forecast look like?**

Seasonal climate forecasts are predictions of how the weather will evolve over the next season (typically three to six months ahead). Day-to-day forecasts are unreliable over such long horizons, so forecasts are instead used to say whether the next season will, on average, show broad differences to normal. Forecasts are therefore usually given as the probability of falling into one of three terciles: below normal, normal or above normal. The statistical fish model uses terciles to summarise whether migration timing is likely to be early, normal or late relative to normal. Instead of terciles, a binary classification was used to summarise water quality predictions in Norway, with the probability of being in two WFD-relevant classes (e.g. above or below 'Good' ecological status).

Quantification and effective communication of forecast quality is a crucial element of seasonal forecasting. Following literature recommendations (Taylor et al., 2015; Gill, 2008), two kinds of forecast quality information were provided alongside forecasts:

1. **Predictability of the future state of the environment.** SEAS5 has 50 ensemble members, or 25 in hindcast mode. Each member is *a priori* equally likely and was used to produce 25 impact model forecasts. The divergence of members provides information about future predictability, with better agreement between members meaning higher predictability. In practice, this information is given as the probability of the tercile, i.e. the percentage of ensemble members which fall into each tercile.

2. **Historic skill.** This describes how well the forecast performed historically when compared to observations. Forecasts should not be used to inform management if the system has no skill, regardless of the agreement between ensemble members. Skill was quantified for each season and tercile using the probabilistic Relative Operating Characteristic Skill Score (ROCSS). This is a simple and easy-to-interpret measure of skill, which is well-suited to communication with decision makers. It ranges from 1 (a perfect forecast) to -1 (a perfectly bad forecast). A value of zero indicates no skill compared to a climatological prediction. A significance test was carried out to indicate whether forecasts were significantly better than climatology ($\alpha = 0.05$). In Norway, only two classes were forecast by the BN and so Matthew's correlation coefficient (MCC) was used instead of ROCSS, as it is well suited to summarizing the overall skill of binary classifiers. MCC ranges between 0 and 1.

To help design forecasting tools which presented these two sources of quality information in a user-friendly way, we held a workshop on visualising and communicating uncertainty. After providing background on the two kinds of forecast quality information, we discussed end user preferences for how the information should be presented. We took as starting point findings from the EU FP7 EUPORIAS project, which had a particular focus on communicating forecast quality (Taylor et al., 2016), including recommendations that: (1) forecasts should not be provided when there is no skill, as research has shown that end users tend to be influenced by the forecast even if it has no value; (2) qualitative skill and uncertainty categories and visual cues should be provided, to help users make sense of skill information; (3) attempts to classify skill as

'good' or 'poor' are subjective, and so thresholds should be decided on together with end users; (4) a single measure of confidence should be considered, that combines quantile likelihoods with a measure of historic skill, to simplify interpretation and to ensure, for example, that managers are mistrustful of forecasts with no historic skill, regardless of whether tercile probability is 'High'; (5) a tiered or layered approach to presenting forecast quality information is a useful means of avoiding confusion, where different levels of information may be selected by different user groups.

Following on from discussions at this workshop, forecast presentation varied somewhat between sites, given a range of preferences. At all sites, however, the tercile probabilities and historic skill scores were categorised and accompanied by descriptive text summaries, to aid interpretation. Managers were involved in deciding on appropriate categories and wording. Tercile probabilities (i.e. agreement between ensemble embers) were split into four categories: 'Very low' (<35%), 'Low' (35-49%), 'Medium' (50-64%) and 'High' (65-100%). For historic skill, the ROCSS text summary was 'Skilled' or 'None', according to whether ROCSS was significantly positive or not. Historic skill given by MCC in Norway was summarized qualitatively as 'None' (< 0.2), 'Low' (0.2-0.39), 'Medium' (0.4-0.59) or 'High' (> 0.6). A combined confidence score was also provided, integrating the two types of forecast quality information. We opted to derive this by setting it to be the same as the tercile probability unless the historic skill was 'None', in which case it was also 'None'. For water quality forecasts in Norway, if class probability was 'High' then overall confidence was the same as the historic skill; if class probability was 'Medium', overall confidence was historic skill reduced by one class. An example of a forecast can be seen for Lake Vansjø at https://watexr.data.niva.no/ (last accessed January 2022).

## 2.5.    Identifying windows of opportunity

'*Windows of opportunity*' for seasonal forecasting were required for the second assessment exercise (Sect. 3.2). These were identified at each site using historic skill scores (Sect. 2.4). SEAS5 hindcasts were compared to ERA5, and impact model forecasts were compared to observations. Skill was calculated for every season in the 24 year period 1993-2016, or longer where possible (1981-2019 for the BN in Norway and 1993-2019 for fish migration timing in Ireland). At the Irish and Norwegian sites, real observations were used to assess skill. Elsewhere, forecasts were compared to 'pseudo-observations', model output derived by running models forced with ERA5 data. Skill calculated using pseudo-observations ignores impact model error and is therefore a best case estimate, although as seasonal climate skill is likely the largest source of uncertainty, this is still a useful first assessment of forecast performance. Statistical significance (95% confidence) of ROCSS was then used to identify windows of opportunity, i.e. season/variable/tercile combinations for which forecast performance was significantly better than expected from a forecast with no discriminative skill (ROCSS = 0). Windows of opportunity reported in this paper summarise those already reported in Mercado-Bettín et al. (2021) for Sau and Mount Bold and for streamflow and lake water temperature forecasting in the Vansjø, as well as including updated results for Wupper using an improved model calibration, and new results for Burrishoole and for lake water quality/ecology forecasting in Vansjø. The windows of opportunity identified present a useful first indication of where seasonal forecasts may be reliable enough to

support decision making, but should be interpreted with some caution due to the small sample size (a short hindcast period is split into 3 terciles, i.e. 8 data points per tercile), the use of pseudo-observations for some sites or variables, and the somewhat subjective 5% significance threshold chosen to identify robust forecasting windows.

## 3. Results

### 3.1. Usefulness of seasonal forecasts during a historic season

In the first assessment exercise, water managers were asked to choose a historic season when seasonal climate resulted in problems in their study site. The events chosen are summarised in Table 2, along with associated surface water impacts and opportunities identified for mitigating the impacts, given a reliable-enough forecast. Dry and hot seasons were chosen in Mount Bold and Wupper, with associated problems with low reservoir water levels, problems meeting demand and poor water quality. A dry season was also selected in Burrishoole, which was accompanied by a later than normal salmon run. Prolonged wet periods and associated lake flooding and poor water quality were selected in Sau and Vansjø.

**Table 2. Seasonal events selected by water managers, associated surface water impacts and management opportunities had advance warning been available at the time.**

| Site | Climate event | Surface water impacts | Management opportunities |
|------|---------------|----------------------|--------------------------|
| Mt Bold reservoir (Australia) | Boreal autumn 2006 (Australian spring; Sep-Nov). A dry and hot autumn during the 'Millennium Drought' (1996 to mid-2010) | High water demand. Low reservoir level at the beginning of the pumping season. Poor water quality. | Strategic planning of water pumping and associated lower water pumping costs. |
| Wupper reservoir (Germany) | Summer 2003 heatwave | Reservoir level had to be lowered substantially to supply drinking water further downstream. Associated eutrophication. Downstream water quality was also impacted. | Store extra water in a series of upstream reservoirs in advance. |
| Burrishoole catchment (Ireland) | Low rainfall in spring 2010, following a very cold winter | Around 80% of salmon migrated during 19th-22nd May, later than average. | Being prepared for sampling (sufficient staff, equipment) during key migration periods ensures efficient data collection and minimizes the impacts of sampling on fish health (e.g. reduced time in traps). |
| Lake Vansjø (Norway) | Very high rainfall in autumn 2000 | High water level, flooding of farm land and sewage stations, high nutrient inputs. Toxic algal blooms in summer 2001 (and proceeding summers until 2007). | Lower lake level in advance. Extra monitoring to screen for toxic blooms at bathing sites. |
| Sau reservoir (Spain) | High precipitation in autumn 2019 | Large water, sediment and organic matter fluxes from upstream, lake flooding, poor reservoir water quality. Increased treatment costs. | Lower the lake level in advance. Store good quality water in an upstream reservoir. |

### 3.1.1    Forecasts for the selected events

Forecasts produced for the seasons of interest and presented to water managers are summarised in Table 3 (see Sect. 2.4 for an explanation of the confidence information that accompanied the forecasts). For climate forecasts, overall confidence in predictions was uniformly low. Even when there was good agreement between forecast ensemble members and therefore high tercile probability, low historic skill and non-significant ROCSS meant that no confidence could be placed in forecasts. However, some positive ROCSS were present (e.g. in Australia) and, although not significant, may be providing added value to freshwater impact model forecasts.

Freshwater impact model forecasts, meanwhile, had medium or high skill in one or more of the variables of interest at most sites, suggesting a lack of sensitivity to seasonal climate (discussed further in Sect. 4).

**Table 3. Summary of forecasts for the historic events selected by water managers.** Abbreviations: cyano: cyanobacteria, D: day of migration, BT: bottom water temperature, P: precipitation, Q: inflow discharge, ST: surface water temperature, T: air temperature, var: variable.

| Site | Target season | Model | Var | Observed tercile | Forecast tercile | Probability[a] | Confidence Skill[b] | Overall confidence[c] |
|---|---|---|---|---|---|---|---|---|
| Mt Bold (Au) | 2006, southern spring (Sep-Nov) | SEAS5 | P | below | below | Medium (60%) | None (0.27) | None |
| | | | T | above | above | Low (40%) | None (0.33) | None |
| | | Impact | Q | below | below | High (72%) | Skilful (0.48) | Medium |
| | | | ST | normal | above | Low (44%) | None (−0.04) | None |
| | | | BT | above | below/normal | Low (36%) | None (0.38/0.4) | None |
| Wupper (Ge) | 2003, summer (Jun-Aug) | SEAS5 | P | below | above | Medium (55%) | None (−0.14) | None |
| | | | T | above | above/below | Low (35/35%) | None (−0.61) | None |
| | | Impact | Q | below | above | Medium (52%) | None (−0.59) | None |
| | | | ST | above | below | Low (44%) | None (−0.15) | None |
| | | | BT | above | above | High (92%) | Skilful (0.71) | High |
| Burrishoole (Ir) | 2010, spring (Mar-Jun) | SEAS5 | P | below | below | Low (48%) | None (−0.23) | None |
| | | | T | normal | below | Low (44%) | None (0.23) | None |
| | | Impact | Salmon mean D | later | later | High (88%) | None (0.25) | None |
| Vansjø (No) | 2000, autumn (Sep-Nov) | SEAS5 | P | above | normal | Low (36%) | None (0.25) | None |
| | | | T | above | below | Medium (56%) | None (−0.26) | None |
| | | Impact | Q | above | below | Low (36%) | None (−0.01) | None |
| | 2001, summer (May-Oct) | Impact | Chl-a | ≤ Poor | ≤ Poor | N/A | High (0.71) | Medium |
| | | | Cyano | ≥ Good[d] | ≤ Moderate | Medium (64%) | High (0.78) | Medium |
| Sau (Sp) | 2019, Autumn (Sep-Nov) | SEAS5 | P | above | above | Low (48%) | None (0.1) | None |
| | | | T | normal | above | Medium (52%) | None (0.17) | None |
| | | Impact | Q | above | above | Low (43%) | Skilful (0.47) | Medium |
| | | | ST | above | normal | High (76%) | None (0.05) | None |
| | | | BT | normal | above | High (100%) | Skilful (0.54) | High |

[a] Probability of the most likely tercile, discretized into categories Low (33-49%), Medium (50-64%) or High (65-100%).
[b] Historic skill score is ROCSS (summarised qualitatively as 'None' for non-significant results, otherwise 'Skilful') or MCC in Norway (discretized into None (< 0.2), Low (0.2-0.39), Medium (0.4-0.59) or High (> 0.6).
[c] Probability and skill were combined into a single score with four classes, None, Low, Medium or High (see Sect. 2.4).
[d] Cyanobacterial blooms did occur near the bathing beaches, but not at the lake monitoring point.

### 3.1.2    Value of forecasts for decision making

Water managers were then asked to assess whether forecasts would have been useful had they been available in advance and, if so, how. Questions and full responses are given in SI2 (see Sect. 6) and are summarised in Table 4, where common themes which emerged across study sites have been highlighted. Managers at all sites could see the potential value of forecasts. However, even given skilful forecasts for at least some of the variables of interest, forecasts would only have been used qualitatively as a pointer to the best strategies, rather than directly feeding into operational management. Main barriers were forecast skill and uncertainty and more general issues of trust. Even where skill was high, water managers said that they would need to observe the forecasts performing well themselves to build confidence that they were providing trustworthy additional information, showing the importance of personal experience. Managers at all but the Norwegian and Irish sites also stated that their trust in the freshwater impact model forecasts was low in part because of the low skill of the seasonal climate forecasts.

**Table 4. Aggregation of water manager feedback on the usefulness of forecasts for the chosen historic seasons.**

| Question | Response | Mt Bold (Au) | Wupper (Ge) | Burrishoole (Ir) | Vansjø (No) | Sau (Sp) |
|---|---|:---:|:---:|:---:|:---:|:---:|
| Would the forecasts have been useful? | Yes | | | | | |
| | Somewhat | ✓ | ✓ | ✓ | ✓ | ✓ |
| | No | | | | | |
| If so, how would they have been used? | Indication of appropriate reservoir management strategies | ✓ | ✓ | | | ✓ |
| | Inform staffing levels/monitoring | | | ✓ | ✓ | |
| Key barriers? | Uncertainty and low historic skill | ✓ | ✓ | ✓ | ✓ | ✓ |
| | Need to develop personal experience of 'added value' | | ✓ | | ✓ | ✓ |

### 3.2.    Windows of opportunity and assessment of their usefulness

In the second exercise for exploring the potential for forecasts to support management, we carried out a more comprehensive assessment of whether the seasons, variables and terciles which could be forecast with reasonable confidence (the windows of opportunity) were considered useful for water management, as well as which windows managers most wished to obtain skilful forecasts for.

There were few windows of opportunity in seasonal climate (Table A1). There were no windows in Ireland, 3 in Germany, 5 in Spain, 9 in Australia and 10 in Norway. The 5% significance level used means that some of these may be false positives

(tests were carried out on up to 108 data slices per site; 4 seasons × 3 terciles × up to 9 variables, so we would expect on average 5 false positives per site).

A substantially larger number of impact model variables showed significant skill (Table 5). This suggests that antecedent conditions/inertia are responsible for much of the skill, further supported by the fact that bottom water temperature was better predicted than surface water temperature (15 versus 5 windows, respectively; Table 5), likely because of its lower sensitivity to seasonal climate (discussed in Sect. 4.2).

Opinions on the usefulness of the windows of opportunity are summarised in Table 5 (full responses are given in SI3; see Sect. 6), together with the windows which water managers were most interested in. All the windows of opportunity for discharge were thought to be of medium or high relevance, and almost all combinations of season/tercile were highlighted as being desirable for management. For surface water temperature, spring to autumn 'above normal' forecasts were seen to be the most useful, due to often strong links between warm summer water and problematic algal growth. Many of the other windows of opportunity in surface water temperature were thought to be of medium or low relevance. As mentioned above, bottom water temperature was the variable that was most successfully forecast, with 15 windows of opportunity across the case study sites. However, it was also the variable that was thought to be least useful for management, with four of the windows being ranked as having low or no relevance. Overall, we found a mismatch between the variables that were thought to be most useful for management, and those which could best be forecast. This can be seen, for example, in the difference between the number of current versus desired skilful windows for the different variables (Table 5; discussed further in Sections 4.1 and 4.2).

In addition, a number of water managers commented that they would require information on more than just the most probable tercile, but rather the likelihood of extremes, which are particularly challenging for management.

Responses from the Irish site are not shown in Table 5 as a large range of population statistics were explored. Three windows of opportunity were found: early median day of migration for trout, later than normal day when 5% of salmon have migrated, and later than normal day when 25% of eel have migrated. These were all thought to be extremely relevant for management. The most desired windows of opportunity were the day when 25% of the population has migrated, the mean day of migration and, for eel, the day when 75% of the population has migrated, although skill in the timing of all percentiles was of interest to check forecast consistency. Although all terciles were thought to be relevant, "earlier than normal" was considered the most useful, as acting on a wrong "early" forecast would have relatively minor consequences, whilst delaying action because of a "later than normal" forecast could result in fish mortality if the forecast were wrong.

**Table 5.** Windows of opportunity in surface water variables and end user assessment of their relevance for management (L: low/none, M: medium, H: high). Ticks show the windows which managers particularly wanted skilful forecasts for. Where letters and ticks coincide, the window was both skilfully forecast and particularly useful. Irish case study responses are given in the text.

| Variable | Boreal season | Tercile or class | Windows of opportunity & their relevance for management | | | | Total windows | Desired windows |
|---|---|---|---|---|---|---|---|---|
| | | | Sau (Sp) | Wupper (Ge) | Vansjø (No) | Mt Bold (Aus) | | |
| Discharge | winter | above | | ✓ | ✓ | ✓ | 6 | 39 |
| | | normal | M | ✓ | ✓ | ✓ | | |
| | | below | | ✓ | ✓ | ✓ | | |
| | spring | above | | ✓ | M ✓ | ✓ | | |
| | | normal | | ✓ | ✓ | ✓ | | |
| | | below | ✓ | ✓ | M ✓ | H ✓ | | |
| | summer | above | | ✓ | ✓ | ✓ | | |
| | | normal | | ✓ | ✓ | ✓ | | |
| | | below | M | ✓ | ✓ | ✓ | | |
| | autumn | above | H ✓ | ✓ | ✓ | ✓ | | |
| | | normal | | ✓ | ✓ | ✓ | | |
| | | below | ✓ | ✓ | ✓ | ✓ | | |
| Surface water temperature | winter | above | | | | | 5 | 15 |
| | | normal | | | | | | |
| | | below | | | L | | | |
| | spring | above | ✓ | ✓ | M ✓ | | | |
| | | normal | | | ✓ | | | |
| | | below | | | M ✓ | | | |
| | summer | above | H ✓ | ✓ | ✓ | | | |
| | | normal | | | ✓ | | | |
| | | below | L | | ✓ | | | |
| | autumn | above | ✓ | ✓ | ✓ | | | |
| | | normal | | | ✓ | | | |
| | | below | | | ✓ | | | |
| Bottom water temperature | winter | above | | | L | ✓ | 15 | 8 |
| | | normal | | | | | | |
| | | below | | | L | M | | |
| | spring | above | L | M ✓ | M | | | |
| | | normal | | | | | | |
| | | below | | M ✓ | M | | | |
| | summer | above | M | M ✓ | ✓ | | | |
| | | normal | | | ✓ | | | |
| | | below | L | M ✓ | ✓ | M | | |
| | autumn | above | L | | | | | |
| | | normal | | | | | | |
| | | below | L | | | | | |
| chl-a cyanobacteria colour total P | Growing season (May-Oct) | upper or lower | N/A | N/A | H ✓ H ✓ M ✓ | N/A | 3 | 3 |

## 4. Discussion: opportunities and barriers for seasonal forecasting to inform water management

### 4.1. Water manager views on forecast value and key barriers

Water managers were generally enthusiastic about the forecasting tools developed and their potential to assist them in preparing for the coming season. They identified actions that could be taken, given a reliable-enough forecast, to help reduce the negative impacts of otherwise unforeseen events. They were well aware of the limited skill of many of the forecasted variables, and were generally comfortable with the idea of working with probabilistic forecasts. For most sites, the act of setting up the impact models was in itself a valuable process, and managers were often enthusiastic about the new system knowledge gained in doing so, and for the workflows to be more generally useful (for example for forecasting at shorter time scales).

Despite general enthusiasm, no-one felt that forecasts could be incorporated directly into operational management straight away. In all cases, forecasts would only be used qualitatively in the first instance, to provide a general indication of how conditions might evolve, rather than to drive operational models (Sect. 3.1.2), matching the findings of Bruno Soares et al. (2018). Trust and lack of personal experience were key issues raised by most managers. The other key limitation, raised at all sites, was forecast quality (i.e. high forecast uncertainty and low skill; see Section 2.4). The similarity in responses across the contrasting study sites suggests these manager viewpoints are likely to be more widely applicable. Forecast uncertainty was a barrier at most sites, as there was often close to an even distribution of probabilities across terciles for the selected historic events (Section 3.1.1), making it difficult for managers to know whether to act based on forecasts. Whilst a reduction in forecast uncertainty (i.e. increased sharpness) could therefore help in increasing uptake of seasonal forecasts in operational water management, studies have shown that even highly uncertain forecasts can significantly improve reservoir management, as long as forecast uncertainty is explicitly accounted for, for example within an optimization framework (e.g. Ficchì et al., 2016). More widespread use of optimization routines in water management could also therefore increase the value of seasonal forecasts for management. Increasing the skill of forecasting systems remains however a top priority (discussed further in Section 4.4), as no matter how sharp the forecasts, they should not be used to inform management if the forecasting system has been proven to have no skill.

There was general enthusiasm for many of the 'windows of opportunity', the variables, seasons and terciles for which the forecasting systems showed most potential. However, there was often a mismatch between what could best be predicted and what was considered most useful (Table 5). Seasonal discharge forecasts were of particular interest, for example, and yet there were few discharge windows of opportunity. Bottom water temperature, meanwhile, could be forecast reasonably well at many sites, and yet had limited management relevance.

At all sites, the sustained uptake of project outputs into the future requires operationalization and maintenance, which is often a challenge as ongoing funding is required. An additional barrier to direct uptake of the tools is that they do not include reservoir operations or interactions between management choices and catchment/lake conditions. However, at many sites there are plans for the knowledge and workflows generated to be used beyond the project duration. In Sau Reservoir in Spain, for example, researchers are now providing monthly reports to the end user for seasonal forecasts with a one month

lead time, whilst in Wupper Reservoir in Germany, Wupperverband are incorporating the workflow, and forecasts derived from it using a variety of climate products, into their longer-term management plans. At the Irish site, parts of the workflow (published in the fishcastr R package developed during WATExR; see Sect. 6) are already being used for other ongoing analyses of fish migration, but additional funding is required for operationalization of the forecasting system and, ideally, expansion to a wider area. At Mount Bold reservoir, SA Water were encouraged by the WATExR project results, and have

decided to invest in an internal follow-on project to establish a seasonal forecasting methodology.

### 4.2.    Sources of seasonal predictability and management implications

As mentioned in the introduction, seasonal predictability in freshwater impact model predictions derives from knowledge of initial conditions and of climate over the target season. At the majority of study sites there were a number of windows of opportunity where freshwater variables could be forecast with reasonable skill (Sect. 3.2). Seasonal climate forecasts

themselves had very limited skill at these extratropical latitudes, so it seems likely that the impact model windows of opportunity were primarily due to models capturing how initial conditions and system inertia influence the target season. This could explain the better performance of bottom water temperature forecasts compared to surface water temperature and discharge forecasts, as the latter two are more sensitive to seasonal climate. Although further work is needed to confirm the sensitivity of the different variables to seasonal climate and initial conditions (and will be the topic of an upcoming paper),

the initial indication is that those variables that are most sensitive to climate over the target season are the hardest to generate reliable seasonal forecasts for (due to low seasonal climate model skill in our study areas), and yet in this case were the variables which were most useful for management, given the importance of streamflow for reservoir operations and of surface water temperature in controlling algal blooms.

The low skill of the seasonal climate forecasts is typical of skill over much of Europe, parts of North America, Russia,

northern China and other mid-latitude areas (Johnson et al., 2019; Maclachlan et al., 2015). Seasonal forecasting to support water management in these areas with low climate model skill will therefore be largely reliant on initial conditions as the main source of seasonal predictability. High quality forecasts, which can be used to inform management, are then most likely in catchments/lakes where initial conditions exert a larger influence. This is the case in larger systems and for variables or ecological species which are less sensitive to seasonal climate. For streamflow forecasts, for example, initial conditions

provide much of the forecasting skill and predictability is highest in slower-responding catchments with larger water storage and groundwater contributions (Pechlivanidis et al., 2020; Girons Lopez et al., 2021; Donegan et al., 2020; Harrigan et al.,

2018). Many successful streamflow forecasting systems do not include seasonal climate model forecasts, showing that a great deal can be achieved using only historic information, in some cases/seasons outperforming predictions which use seasonal climate model forecasts (e.g. Peñuela et al., 2020; Arnal et al., 2018). In lakes and reservoirs, the storage buffering

effect has also been shown to reduce the importance of streamflow forecast skill, particularly when the reservoir capacity is large compared to variability in inflow (Maurer and Lettenmaier, 2004; Turner et al., 2017). Studies looking at sources of predictability for seasonal water quality and ecology are currently lacking, but similar concepts will likely hold.

## 4.3. Do seasonal climate forecasts provide added value at extratropical latitudes?

Where seasonal climate forecasts are skilful, they undoubtedly have potential to provide added value to surface water impact

model forecasts. All water managers in this study were very enthusiastic about the potential benefit of skilful climate forecasts for improving their operations and were particularly interested in variables which are more likely to be sensitive to seasonal climate (e.g. discharge and surface water temperature). Several studies have shown that skilful seasonal climate forecasts can lead to sometimes large improvements in streamflow forecasting ability (e.g. Shukla and Lettenmaier, 2011), which may, for example, have economic value for reservoir operations (Maurer and Lettenmaier, 2004; Turner et al., 2017).

However, in our study catchments, seasonal climate models did not produce skilful forecasts for the selected historic events (Sect. 3.1), and there were few windows of opportunity for seasonal climate (Sect. 3.2). In areas where seasonal climate model skill is low, the extra resources required to work with seasonal climate data may not be worth potentially marginal performance gains, particularly as poor seasonal climate forecasting skill may reduce trust in any skilful impact model forecasts that use seasonal climate as input (Sect. 3.1.2). Particularly in larger catchments and lakes, which are less sensitive

to seasonal climate, it is likely that attention would be better spent on developing simpler benchmark systems. Methods inspired by Ensemble Streamflow Prediction are likely candidates, potentially made more nuanced by, for example, using North Atlantic Oscillation (NAO) index or other climate signals to condition the forecast (e.g. Donegan et al., 2020; Najafi et al., 2012; Sabzipour et al., 2020), or longer-term climate projections (Gronewold et al., 2017).

In systems that are particularly sensitive to meteorological forcing at seasonal timescales (e.g. small catchments and

lakes/reservoirs with short residence times), and where the benefits of any windows of opportunity are large (e.g. a large potential cost saving or particularly sensitive drinking water source/habitat), then the potential benefits of incorporating seasonal climate data are greater. In this case, it may be worth incorporating seasonal climate data into the forecasting workflow, as long as there are some windows of opportunity at lead times of interest for management.

## 4.4. Future priorities for more skilful seasonal predictions

A key barrier to the use of seasonal forecasts in operational management is forecast performance, in particular poor historic skill (Sect. 4.1). To help improve performance, we see the need for progress to be made on two fronts:

(1) *Improvements in seasonal climate model skill*. Seasonal climate models are under active development, and recent advances in the prediction of climate teleconnections, such as the NAO in Europe (Wang et al., 2017; Scaife et al., 2014; Svensson et al., 2015), may lead to improvements in coming years.

(2) *Improvements in impact model performance*. Improvements in impact models themselves may also be required, as in some cases errors derived from impact models may be the dominant source of uncertainty (e.g. Cho et al., 2016). However, probably the greatest potential here is through improved/increased data collection. Although not considered in detail here, the importance of this cannot be understated. For example, observed data is fundamental to developing and calibrating trustworthy impact models, correctly initialising impact models and evaluating the historic skill of forecasting systems, 495 which is a key element for building trust in predictions.

## 5. Conclusions

In this study, we have explored whether pilot seasonal forecasting tools developed at five case study sites could usefully support practical water management. Tools integrated seasonal climate model forecasts and freshwater impact models to produce forecasts of streamflow, lake water level, lake water temperature and, at some sites, lake water quality/ecology and 500 fish migration timing. Co-development was a key part of the process, i.e. researchers and end users worked closely to design tools that were relevant and tailored to the individual needs at each of the study sites. This meant that the user-community was able to make well-informed assessments of forecast skill and qualitative value for decision-making. Key outcomes include:

- At the majority of case study sites there were windows of opportunity where surface water forecasts could be produced
with enough skill to be potentially useful for management.

- End users were enthusiastic about the potential for improved decision making and identified actions that could be taken based on forecasts. However, even skilful forecasts would only be used qualitatively in the first instance, until trust had been built up through practical hands-on experience.

- Reduced uncertainty and higher historic skill were identified as key requirements for the operational use of forecasts, as
was an ability to forecast more extreme seasonal events than just terciles (below normal, normal or above normal). Improved procedures within operational water management that take into account uncertain forecasts (e.g. optimization) would also likely result in an increase in forecast management value.

- Where seasonal climate forecasts are skilful, they undoubtedly have potential to provide added value to freshwater model forecasts and assist management, in particular in smaller systems which are more responsive to climate.

- Outside the tropics, seasonal climate forecast skill is limited. Despite this, forecasting within the water sector can still be usefully carried out, but relies on seasonal predictability derived from antecedent/initial conditions and system inertia. The best chance of developing useful seasonal forecasting tools is then in slower-responding systems (e.g. larger

catchments and lakes), which are less sensitive to climate over the target season. In this case, time is probably best spent on developing tools which use re-sampled historical meteorological data rather than seasonal climate model output to force impact models.

- Seasonal climate model forecasts with only patchy skill are most likely to be worth incorporating into freshwater seasonal forecasting workflows when: (1) the system is particularly sensitive to seasonal climate (e.g. small catchments and lakes), and (2) the potential benefits of any windows of opportunity are large.

## 6. Supplementary information, supporting data, tools and code

Supplementary data (SI1-3) are available at https://doi.org/10.5281/zenodo.5906258 (Jackson-Blake, 2022). Seasonal forecasting tools and/or underlying code are available for several of the study sites in the WATExR GitHub repository (website: https://nivanorge.github.io/seasonal_forecasting_watexr/; repository: https://github.com/NIVANorge/seasonal_forecasting_watexr; last accessed Jan 2022). The fishcastr R package, for seasonal forecasting of timing of diadromous fish migration, is publicly available (French, 2021).

## Appendix

| **Box A1.** Questions posed to water managers to assess the usefulness of the windows of opportunity. |
| --- |
| Part I: Survey of whether each window of opportunity is useful: <br><br> 1. The model has skill in forecasting $x_{ij}$. How relevant is this for you for management? (Where $x_{ij}$ = the window of opportunity $x$ for season/tercile $j$ and variable $i$; question repeated for each $j$) <br><br>     Answer: Multiple choice, options: extremely relevant, somewhat relevant, not relevant <br><br> 2. If any of the windows of opportunity for this variable are relevant to management, please comment on how they might be useful. <br><br>     Answer: free text <br><br> Questions 1 and 2 were repeated for each variable, if windows of opportunity were present <br><br> Part II: Profiling of which variables, seasons and terciles skilful predictions are most desirable for: <br><br> 1. For variable $x$, which season(s) and tercile(s) would be most useful for you to have skilful seasonal forecasts for to support management (if any)? (Where $x$ = simulated variable) <br><br>     Answer: checkbox grid, with one row per season and one column per tercile <br><br> 2. Please describe any variables that weren't included and which are of interest to you, and for which season(s) you would want predictions. |

**Table A1. Climate variables, seasons and terciles for which SEAS5 had significant skill, as assessed by comparison to ERA5 over the period 1993-2016.**

| Site | Number of skilful /total combinations[a] | Skilful climate variable/season/tercile combinations | | |
|---|---|---|---|---|
| | | **Boreal season** | **Variable[b]** | **Tercile** |
| Mount Bold, Australia | 9/108 | Spring (Mar-May) | rlds | above |
| | | Autumn (Sep-Nov) | cc | above |
| | | | petH | above |
| | | | psl | above |
| | | | rsds | above |
| | | | tas | normal |
| | | Winter (Dec-Feb) | psl | above |
| | | | tdps | above |
| | | Summer (June-Aug) | cc | normal |
| Wupper, Germany | 3/96 | Spring (Mar-May) | tdps | above |
| | | Winter (Dec-Feb) | rlds | normal |
| | | | vas | below |
| Burrishoole, Ireland | 0/18 | None | | |
| Vansjø, Norway | 10/96 | Spring (Mar-May) | psl | normal |
| | | | psl | above |
| | | | tas | above |
| | | | tcc | normal |
| | | | tdps | below |
| | | | uas | above |
| | | | vas | below |
| | | Winter (Dec-Feb) | rlds | normal |
| | | | rsds | above |
| | | | tcc | below |
| | 2/108 | Early summer (May-Jul) | rsds | above |
| | | Autumn (Sep-Nov) | cc | above |
| Sau, Spain | 5/108 | Spring (Mar-May) | cc | above |
| | | | psl | above |
| | | Summer (Jun-Aug) | cc | above |
| | | | tdps | above |

[a] ROCSS were calculated for z total data 'slices', where $z = x$ met variables $* y$ seasons $* 3$ terciles.

[b] Meteorological variable abbreviations: psl: surface pressure, tcc: total cloud cover, uas: 10 m $u$ wind component, vas: 10 m $v$ wind component, tas: 2 m temperature, tdps: 2 m dewpoint temperature, rsds: downwards surface solar radiation, rlds: downwards surface thermal radiation, tp: total precipitation.

**Author contribution**

AF, DMB, FC, MDF, MS, LJB, RM and TM developed and applied the modelling workflows, with input from all co-authors; LP facilitated the first assessment exercise; LJB and FC conducted the second assessment exercise; LJB prepared the manuscript with contributions from all co-authors.

**Competing interests**

The authors declare that they have no conflict of interest.

**Acknowledgements**

Thanks to all our co-developers, some of whom are co-authors, but also: Carina Isdahl (Morsa, Norway), Juan Carlos Garcia Pradell (ATL, Spain) and Marc Scheibel, Eleni Teneketzi and Paula Lorza (Wupperverband, Germany). Thanks to Sixto Herrera, Justin Brookes, Sigrid Haande, Francesca Pianosi and an anonymous reviewer for valuable contributions and/or comments on the manuscript. Data was provided by ATL, SA Water, Wupperverband, NIVA, NVE and the Marine Institute. SEAS5 and ERA5 meteorological data were based on data and products of the European Centre for Medium-Range Weather Forecasts (ECMWF). Funding: This work was carried out as part of the WATExR project, part of ERA4CS, an ERA-NET initiated by JPI Climate and funded by MINECO-AEI (ES, projects PCIN-2017-062 and PCIN-2017-092), FORMAS (SE), BMBF (DE), EPA (IE), RCN (NO, project 274208), and IFD (DK), with co-funding by the European Union (Grant 690462).

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
