# Peer review of "Opportunities for seasonal forecasting to support water management outside the tropics"

_Hydrology and Earth System Sciences, 2021_

## Author Response (AR1)

Many thanks to the two reviewers for a very constructive and useful set of comments. Responses to the comments are available in the 'Discussions' history on HESSD. Below, we have just documented the changes made to the manuscript as part of the first revision.

**Reviewer 1**
This paper presents the findings of a research project into the ability to use seasonal forecasting within water management. This is a topic of great value to water management as highlighted in the introduction. This is a well-written and presented paper, with it being written such that it is easily accessible by non-specialists. I appreciate this paper has a more technical counterpart (Mercado-Bettín et al 2021), however, this paper is lacking some context and information which would strengthen the reader's understanding. My recommendation is that this paper is published after minor revisions.

Major comments

1. Introduction
1) Introduction paragraph 3 (Lines 54- 70). This is a key paragraph that needs expanding upon, rather than just stating what products and who used them but how accurate they are. This information should be used in the discussion sections as well with respect to the outcomes of this paper.
Changes made: We have strengthened this paragraph, adding in more information on how successful the streamflow and lake level forecasting products that we mention here are. We have done this only briefly, as the main aim of this paragraph was to point out that streamflow (and to a lesser extent water level) seasonal forecasting has received a considerable amount of attention over the last few decades, whilst seasonal forecasting of water quality and ecology has received little attention, and is therefore the focus of this study. In the discussion, we already mention the skill of streamflow forecasting systems in relation to our results in the last paragraph of Section 4.2 and the first paragraph of Section 4.3, and we have added in a new sentence at the end of Section 4.4 which ties in findings from Cho et al.'s study of seasonal nutrient flux modelling in a Korean catchment.

2. Methods
2) Whilst Figure 1 shows the general location of the sites chosen, no further location details are given. Maps of the catchments should be given with elevation.
Changes made: We have added catchment maps as figures in the supplementary information. Elevation information has been added to Table 1

3) This paper would benefit from more detail of the catchments. A brief description of the catchments land use and how many people the reservoir's supplies.
Changes made: We have added land use information, as well as a little more background data on the lakes/reservoirs, to Table 1. We have added information on the number of people served by drinking water reservoirs to the text, and generally added more information about all the study sites to section 2.1.

2.3. Forecasting work-flows

4) Whilst a more detailed description of the ERA5 and SEAS5 data is within the paper (Mercado-Bettín et al 2021). This paper would benefit from a more detailed ERA5 and

SEAS5 data description. Firstly, which data sets were used and for which model. Secondly, the spatial resolution of both of the data sets should be stated.
Changes made: Added more information (penultimate paragraph of Section 2.3)

5) Further to the comment above more detail on why the Seas5 and ERA5 data were chosen. Is this because SEAS5 is considered to be the best forecast? If so a discussion of this should be presented in the introduction. Similarly, why was ERA5 reanalysis chosen over other potential local sources of data?
Changes made: Added more information (penultimate paragraph of Section 2.3)

4.0 Discussion: opportunities and barriers for seasonal forecasting to inform water management
6) Do you think the current spatial variation of the SEAS5 data played a part in the inaccuracies in the forecasting tool?
Response: I'm not quite sure I understand the comment, that the spatial variation in skill in SEAS5 across Europe affects the surface water forecasting skill? If I have understood the question, then yes perhaps a little, and the extent to which will be quantified in a future study (mentioned in Section 4).

**Reviewer 2**
This is an interesting article, reporting results of a research project on the value of seasonal forecasting for water management across different study sites in Europe. The article is very well structured and enjoyable to read, and I think it makes a good contribution to HESS, where studies on the linking between forecasts-hydrology-water manggagement have become relevant to a growing community. I would thus recommend it for publication after some revisions. Below are some comments and suggestions for improving the manuscript.

[1] Sec 2.2 - The authors mention "water managers" being involved in the design and testing of the tools. Similarly on P. 10 L. 245, the authors mention "stakeholders were asked to choose a historic season ..." It would be useful to give more information about whom specifically was involved: how many people for each study site, and their role and responsibilities in their organisation (and clarify whether "stakeholders" mentioned on P. 10 are the same as the "water manager" in Sec. 2.2). I suppose the term may refer to either technical staff (who is responsible for running models and analysing data, but often does not have direct responsibility to make decisions) or executive managers (who do take decisions but often do not directly analyse data or apply models). Most likely, the views and opinions of these two groups are different, even if they all work in the same organisation, as they have different expertise and different responsibilities (see for example the analysis reported in Höllermann and Evers, 2019). A bit more information about whom specifically was involved in this study would be very useful here to put the results into context.
Changes made: Added more information on the end users (how many people, their roles) to Table 1, as well as more background on the end user organisations in a new paragraph at the start of Section 2.2. For clarity, we have also changed "stakeholder" to "water manager", "manager" or "end user" throughout the text.

[2] P. 6 L. 130: "A workshop on communicating and visualising seasonal forecast uncertainty". What were the outcomes of this workshop? Uncertainty communication and visualisation is a very interesting topic and any new insights would be useful to share. Why not reporting some of the key findings on this topic too?

Changes made: More information on this workshop, and the outcomes of it, have been added to Section 2.4 (and referred to earlier in the text, in Section 2.2).

[3] P. 16 L. 329: "... managers were often enthusiastic about the new system knowledge gained in doing so and for the workflows to be more generally useful". It would be interesting to know more about how the knowledge and workflows generated in this project will be used beyond the project duration. Are forecasting and impact models (or at least, some elements of them) going to be transferred to the water agencies, so that they can keep using them in the future? What challenges did the authors face in such knowledge transfer, and how they plan to overcome barriers to adoption? If models are not going to be directly embedded into the practice of water agencies, have these at least been influenced by the project results, and how? Again, these would be interesting experiences to share. Most research projects in this field produce interesting insights but are rarely followed up by a sustained uptake of the project outputs - some discussion of these problems would be very interesting in my opinion.

Changes made: Added a new paragraph to the bottom of Section 4.2 which briefly discusses some of these very interesting issues. This could be a much bigger topic of discussion!

[4] P. 16 L. 338 "A reduction in uncertainty and higher historic skill are therefore still likely to be general requirements for increased uptake of seasonal forecasts in operational management" and P. 19 L. 417: "Reduced uncertainty and higher historic skill were identified as key requirements for the operational use of forecasts..."

This conclusion may be formulated in a more nuanced way. My experience from being involved in studies (e.g. Penuela-Fernandez et al 2020 and Ficchi et al 2016) where forecasts were directly incorporated into operational decision-making procedures via optimisation, is that the link between forecast skill (how accurate the forecast is in predicting inflows) and forecast value (how useful it is to improve decisions) is quite complex. When using optimisation to generate decisions, the real "game changer" is whether forecast uncertainty is explicitly represented and accounted for (for example through probability distributions or ensembles) or not. If it is, optimisation performances significantly improve and can even approximate performances delivered by "perfect" forecasts (at least for shorter lead times, as shown in Ficchi et al 2016 with 10-days-ahead inflow forecasts). I appreciate that in practical settings optimisation is still relatively unaccepted/unused, and most managers will use forecasts in a qualitative way - i.e. to support their thought process and decision-making, not to feed into optimisation routines. Still, I think it is important to convey the message that forecasts can have value even if their skill is relatively low - as the studies cited above have shown. I think the conclusion that "high historic skill .... is a key requirement for operational use of forecasts" has more to do with gaining trust of users, rather than an "objective" requirement for forecasts to be useful.

Changes made: attempted to make these two parts of the text (Section 4.1 second paragraph; Conclusions) more nuanced, as suggested. Would be interesting to hear whether you agree that a lack of historic skill means that forecasts should not be used, regardless of their uncertainty/sharpness.

[5] P. 17 L. 355 'the initial indication is that those variables that are most sensitive to climate over the target season are the hardest to generate reliable seasonal forecasts for (due to low seasonal climate model skill in our study areas), and yet are also the variables which are most useful for management."

I wonder if this conclusion may be the result of some ambiguity in the answers collected here. When water managers said which forecasts would be more useful, did they think of those variables whose foresight would really be key for better decisions, or did they think of the variables they currently find more difficult to predict? Put it another way, when managers said that certain forecasts are less useful, did they say so because they genuinely do not need to know about those variables, or because they are already able to guess them reasonably well, as they strongly depend on antecedent conditions? If this confusion was present, that may be the (self-evident) reason why "those variables that are most sensitive to climate ... are most useful for management"

Changes made: made changes to the first paragraph of section 4.2. We have generally toned this part of the paper down (e.g. deleted the bullet point relating to this point in the conclusions), as on reflection we think that it was mostly due to the importance of streamflow forecasts for water management, rather than implying that there is a more general rule that variables that are more sensitive to seasonal climate are more useful for management. Perhaps that is the case, but we can't say much about it from the results in this paper.

Minor specific points

p.3 l. 86: " real life management situations" the wording here may suggest that the use of forecasts was tested in real life - for instance to manage an extreme event occurring during the project duration. Studies of this kind are rare but they do exist (see for example Emerton et al 2020). As this is not the case here, and this is a simulation-based study only, it would be worth clarifying.

Changes made: deleted "real life management situations"

P. 8 L. 171: so I understand the algal bloom risk model does not use seasonal weather forecasts but only antecedent conditions. Is that correct? Please clarify

Changes made: clarification added (Section 2.3, 3rd bullet point)

P. 10 L. 241: "5% significance does not necessarily reflect the practical decision-making value of forecasts" Unclear. How is the "practical decision-making value" defined and/or assessed then?

Changes made: re-written, as the point was more that we picked out the windows of opportunity using a fairly arbitrary 5% significance threshold.

Table 2, row 3, the "management opportunity" for Burrishoole site is defined as "Being prepared for data collection during key migration period is very important to reduce fish mortality" This comes a bit unexpected. I am clearly not an expert of fish management, but why being prepared reduces fish mortality? Is data collection harmful to fishes? please clarify

Changes made: Yes, the original was over-stated! Re-written now.

P. 11 L. 260: "Impact model forecasts... suggesting a lack of sensitivity to seasonal climate". This hypothesis could be easily tested by calculating the skill of an ensemble streamflow prediction systems (or equivalent concept for the ecological models). The authors mention this possibility in the Discussion, but I suppose it should be relatively easy to actually run the simulation and calculate the skills, given that all the models and datasets to do so are available?

Response: Yes, a thorough assessment of the origin of the forecasting skills for the windows of opportunity is something we are really interested in. However, we would like to allocate skill separately to warm-up, transition month and target season, meaning a bit

more work is required than just comparing ESP-type forecasts and the seasonal climate runs. These experiments are currently being carried out and results will soon be analysed and written up in a separate paper.